# Different Metabolism and Toxicity of TRANS Fatty Acids, Elaidate and Vaccenate Compared to Cis-Oleate in HepG2 Cells

**DOI:** 10.3390/ijms23137298

**Published:** 2022-06-30

**Authors:** Farkas Sarnyai, Éva Kereszturi, Kitti Szirmai, Judit Mátyási, Johanna Iman Al-Hag, Tamás Csizmadia, Péter Lőw, Péter Szelényi, Viola Tamási, Kinga Tibori, Veronika Zámbó, Blanka Tóth, Miklós Csala

**Affiliations:** 1Department of Molecular Biology, Semmelweis University, H-1094 Budapest, Hungary; sarnyai.farkas@med.semmelweis-univ.hu (F.S.); kereszturi.eva@med.semmelweis-univ.hu (É.K.); szirmaikitti@gmail.com (K.S.); szelenyi.peter@med.semmelweis-univ.hu (P.S.); tamasi.viola@med.semmelweis-univ.hu (V.T.); tibori.kinga@med.semmelweis-univ.hu (K.T.); zambo.veronika@med.semmelweis-univ.hu (V.Z.); 2Department of Inorganic and Analytical Chemistry, Budapest University of Technology and Economics, H-1111 Budapest, Hungary; matyasi.judit@vbk.bme.hu (J.M.); al-hagj@edu.bme.hu (J.I.A.-H.); 3Department of Anatomy, Cell and Developmental Biology, Eötvös Loránd University, H-1053 Budapest, Hungary; tamas.csizmadia@ttk.elte.hu (T.C.); peter.low@ttk.elte.hu (P.L.)

**Keywords:** trans fatty acid, lipotoxicity, obesity, cardiovascular diseases, diabetes, non-alcoholic fatty liver disease, endoplasmic reticulum stress, ceramide, diacylglycerol

## Abstract

Trans fatty acids (TFAs) are not synthesized in the human body but are generally ingested in substantial amounts. The widespread view that TFAs, particularly those of industrial origin, are unhealthy and contribute to obesity, cardiovascular diseases and diabetes is based mostly on in vivo studies, and the underlying molecular mechanisms remain to be elucidated. Here, we used a hepatoma model of palmitate-induced lipotoxicity to compare the metabolism and effects of the representative industrial and ruminant TFAs, elaidate and vaccenate, respectively, with those of cis-oleate. Cellular FAs, triacylglycerols, diacylglycerols and ceramides were quantitated using chromatography, markers of stress and apoptosis were assessed at mRNA and protein levels, ultrastructural changes were examined by electron microscopy and viability was evaluated by MTT assay. While TFAs were just slightly more damaging than oleate when applied alone, they were remarkably less protective against palmitate toxicity in cotreatments. These differences correlated with their diverse incorporation into the accumulating diacylglycerols and ceramides. Our results provide in vitro evidence for the unfavorable metabolic features and potent stress-inducing character of TFAs in comparison with oleate. These findings strengthen the reasoning against dietary trans fat intake, and they can also help us better understand the molecular mechanisms of lipotoxicity.

## 1. Introduction

All the endogenous and most of the ingested fatty acids (FAs) are either fully saturated (SFAs), e.g., palmitate (16:0), or have one or more double bonds of cis configuration, e.g., oleate (18:1 cis-Δ9). This is because humans, like the vast majority of other organisms, synthesize monocarboxylic acids of saturated chains first and create a variability by elongating these chains and/or inserting cis double bonds in them. However, people usually consume considerable amounts of FAs containing at least one double bond in trans configuration, i.e., trans fatty acids (TFAs), and these derive from two origins. The natural source of TFAs is a bacterial isomerization that occurs in the rumen [1]. The predominant ruminant TFA (rTFA), vaccenate (18:1 trans-Δ11) is therefore ingested with the meat and dairy products of ruminant animals. A newer source of TFAs appeared with industrial hardening of plant oils, and this is how elaidate (18:1 trans-Δ9) the main industrial TFA (iTFA) increasingly appears on our plates [2].

Potential health effects of both kinds of trans fats have been debated in the past few decades. They have been implicated in various disorders on the basis of in vivo observations. An association has been found between trans fat intake, all-cause mortality and cardiovascular mortality in a meta-analysis of observational studies, and the dominant role of iTFAs was suggested [3]. Difference between the harmfulness of iTFAs and rTFAs is further supported by the finding that an elaidate-rich diet stimulates atherosclerosis, whereas vaccenate-rich butter protects against it in a mouse model [4]. Meta-analysis of prospective cohort studies combined with retrospective case–control studies revealed a strong association between TFA consumption and the incidence of coronary heart disease at a pooled relative risk of 1.29 [5]. Although limited amounts of TFAs have been suggested to improve life expectances [6], the repeatedly demonstrated correlation with cardiovascular diseases made TFAs widely considered as harmful food components and even led to legislations aiming to eliminate them from food products in several countries. The potential relation between TFA intake and cancer is much less established, though it has also been proposed in a few epidemiological studies [7]. Chronic TFA intake has been also found to be associated with the risk of type 2 diabetes [8], while several in vivo studies indicated neutral or beneficial effects of TFA ingestion and suggested a protective role of vaccenate and trans-palmitoleate (16:1 trans-Δ9) against diabetes [3,9]. A recent cohort study found that diabetes risk was not associated with iTFAs or with trans-palmitoleate but was inversely associated with vaccenate and certain other rTFAs [10].

The in vivo, epidemiological and clinical studies provide convincing evidence that the effects of TFAs deserve attention, and more focus should be given to the underlying mechanisms at cellular and molecular levels. We have recently published our findings regarding the comparison of TFAs with palmitate and oleate in a model of β-cell lipotoxicity [11,12]. Damages inflicted by FA oversupply in various tissues and organs, including adipose tissue, muscle, vasculature, liver, heart and pancreas, play an important role in the development of cardiovascular diseases and diabetes. Under these lipotoxic conditions, the intense FA uptake results in an overwhelming fatty acyl-CoA abundance, which easily saturates the oxidizing efficiency and thus pushes the biosynthetic pathways of lipid metabolism. Triacylglycerols (TGs) are deposited in lipid droplets according to the storage capacity of the affected cell types, but there is limited room for membrane lipids, so their synthesis is jammed, causing the accumulation of intermediates that are normally present at minute concentrations [13]. Ceramides are key intermediates of sphingolipid synthesis. Their de novo generation consumes two fatty acyl-CoA molecules as they are amides of a FA and a sphingosine derived from palmitoyl-CoA and serine [14]. The potential harmfulness of ceramides has been repeatedly demonstrated. They are implicated in the development of cardiovascular diseases and diabetes and are emerging as new biomarkers of these pathologies [15].

In accordance with the literature, cis-oleate was found to be remarkably less deleterious than palmitate, and it also greatly reduced the palmitate-induced damage in our experiments in RINm5F rat insulinoma cells [11,12]. These cells did not show obvious TG deposition; however, ceramides and diacylglycerols (DGs) accumulated, and lipotoxicity was evident from decreased viability, increased apoptosis, the induction of ER stress and the activation of stress-activated protein kinase (SAPK)/c-Jun N-terminal kinase (JNK). Most importantly, both elaidate and vaccenate proved to be as tolerable and as protective as oleate, so our findings did not support either the pronounced harmfulness of TFAs or the difference between iTFAs and rTFAs in this cellular model. The most notable difference we could document between cis and trans unsaturated FAs regarded their impact on DG levels and, particularly, their incorporation into ceramides. TFAs caused a bigger elevation in DG concentrations, and they, unlike oleate, incorporated into ceramides significantly [11,12]. We found this phenomenon worthy of further investigation due to its potential connection with the known long-term health effects.

The aim of the present work was to elucidate the metabolic fate, toxicity and protective potential of elaidate and vaccenate in a cell line relevant for metabolic disorders and capable of TG synthesis. We treated HepG2 human hepatocellular carcinoma cells with BSA-conjugated FAs (palmitate, oleate, elaidate or vaccenate) alone or with the unsaturated ones in combination with palmitate at two different ratios. We assessed the FA profile and the amount of a wide range of lipid intermediates, i.e., DGs, TGs and ceramides in the cells, and examined how the viability, morphology and markers of ER stress and related apoptosis were affected by the different treatments.

Our data provide the first in vitro evidence for some different cellular effects of trans and cis unsaturated FAs. Beside the previously reported incorporation of TFAs into ceramides, here we also demonstrate that a combination of palmitate with one of the TFAs causes more pronounced stress and cell damage and more reduced viability compared with the simultaneous addition of palmitate and oleate. These results strengthen the link between ceramide and DG accumulation and cellular dysfunctions in obesity-related metabolic conditions and may help us understand the epidemiological observations regarding the health effects of trans fat.

## 2. Results

### 2.1. Fatty Acid Treatments and Viability of HepG2 Cells

MTT assay was performed to assess changes in the viability of HepG2 cells after 24 h long incubations with the investigated FAs at various (100, 200, 400 and 800 μM) concentrations (Figure 1a). A strong and concentration-dependent destructive effect of palmitate was detected above 100 μM doses in accordance with the widely reported toxicity of this saturated FA. The unsaturated FAs, regardless of their configuration, only affected cell viability at higher concentrations, and they caused a decrease of approximately 15% even at the highest level, proving that all the three of them are similarly tolerable and markedly less toxic than palmitate when administered alone in these experimental conditions (Figure 1a). None of the investigated FAs were cytotoxic below a 100 μM concentration, and the concentration at which all of them were toxic was 800 μM.

The observed extent of toxicity, in accordance with literature data, makes the 800 μM concentration suitable for the investigation of protective interactions. Thus 800 μM palmitate was combined with 200 μM (i.e., 4:1 ratio) or 400 μM (i.e., 2:1 ratio) unsaturated FA in the combinational treatments. Although the overall FA challenge reached 1000 or 1200 μM concentrations, all six combinations turned out to be less deleterious than palmitate alone (Figure 1b). Oleate protected the cells from palmitate-induced injury the most effectively, its 2:1 combination with palmitate was only mildly damaging. The two TFAs were also protective but significantly less so at a 2:1 ratio, i.e., their ameliorating effect did not grow with their increasing concentration as in case of oleate (Figure 1b).

### 2.2. Utilization of Fatty Acids by HepG2 Cells at Low, Non-Toxic Concentration

The hepatoma cells were first treated with the four investigated FAs at 50 μM concentration for 8 h, and the FA content was analyzed both in the incubation medium and in the cell samples by GC-FID analysis. The reduction of FA concentrations in the medium together with the elevation of cell-associated FA content convincingly show that the cells efficiently take up all the four FAs under the applied experimental conditions. The initial level of each administered FA was approximately halved by the end of the incubations and some cell-derived FAs appeared in the supernatant, but their concentration remained below 5 μM (Appendix A). Palmitate (50 μM) was also added together with an unsaturated FA (12.5 or 25 µM), and the consumption in these combined treatments was similar to what we saw in the single treatments (Appendix A). The FA disappearance in the medium was accompanied by FA buildup in the cells. Increased amount of the supplemented FA could be observed in every treatment, and derivatives of intracellular metabolism were also detected in case of TFAs, i.e., 16:1 trans-Δ7 and 16:1 trans-Δ9 in the elaidate and vaccenate treated cells, respectively (Figure 2, Appendix A). Interestingly, the cellular amount of palmitate increased more efficiently when the saturated FA was added together with any of the unsaturated FAs (Figure 2b, Appendix A).

The amount of some biosynthetic lipid intermediates, i.e., ten TGs (Appendix A), six DGs (Appendix A) and three ceramides (Appendix A), was also assessed in the cell samples by using LC-MS/MS. Palmitate caused a greater increase in these values compared to the unsaturated FAs, and the combinations generally resulted in lipid levels between those induced by palmitate and the unsaturated FAs alone (Table 1). However, the moderate accumulation of TGs was stimulated by the 2:1 palmitate/unsaturated FA mixtures the most effectively (Table 1, Appendix A).

### 2.3. Utilization of Fatty Acids by HepG2 Cells at High, Toxic Concentration

BSA-conjugated FAs were added to HepG2 cells at 800 μM final concentration in single treatments to investigate their cellular effects and metabolism, and the same toxic dose of palmitate was also combined with each unsaturated FA at 4:1 and 2:1 ratios to compare the protective effects. These conditions were selected on the basis of our previous experiences, the results of viability assays (see above) and methodology reported in the relevant literature.

The reduction of the FA content in the incubation medium was assessed by GC-FID analysis. Each administered FA was effectively consumed from the medium, the drop in the concentrations was similar in all cases and more than 25% of the initial value during the 8 h long incubations (Appendix A). The cell-derived FAs appeared again in the supernatant, but their concentration remained below 20 μM (Appendix A).

The adsorption, uptake and utilization of FAs were also assessed in the hepatoma cells by the GC-FID analysis of cellular FA profiles, i.e., the main saturated and mono-unsaturated FAs after 4 h and 8 h long incubations. The measured amounts were normalized to the protein contents of the corresponding samples, and FA ‘concentrations’ were expressed as µg/mg protein. All the four FAs supplemented to the cells in our experiments integrated effectively as indicated by the parallel time-dependent increase in their concentrations (Figure 3, Appendix A).

Rising levels of elongated (stearate, 18:0), desaturated (palmitoleate, 16:1) as well as elongated and desaturated (oleate) derivatives were observed after palmitate treatment, but the palmitate:oleate ratio was still largely increased compared to the control (7.3 vs. 1.3 at 4 h and 11.8 vs. 1.4 at 8 h), and oleate concentration still remained far below that measured in oleate treated cells (31.4 ± 3.2 vs. 237.7 ± 30.3 µg/mg protein at 4 h and 34.8 ± 7.3 vs. 362.6 ± 48.4 µg/mg protein at 8 h) (Appendix A). The intermediates of elaidate and vaccenate degradation, i.e., 16:1 trans-Δ7 and 16:1 trans-Δ9, respectively, appeared at well detectable amounts in the TFA-treated cells, while the level of 16:1 cis-Δ7 did not change upon oleate supplementation (Figure 3, Appendix A).

When palmitate was added together with an unsaturated FA, both of them integrated into the lipidome of the cells, though there are signs of a mutual interference. The intracellular palmitate content reached slightly (at 4 h) or markedly (at 8 h) lower values than at single addition, yet the reduction was statistically not significant (Figure 4 and Appendix A). Similarly, the unsaturated FA contents in these combinational treatments were lower than one half or a quarter of what was caused by twice or four times higher doses of the corresponding unsaturated FA alone.

For instance, oleate supplementation alone increased oleate content from 24.9 ± 3.3 (control) to 362.6 ± 48.4 µg/mg protein after 8 h, which is a 337.7 µg/mg protein increase, while half that much oleate added together with palmitate increased oleate content by 84.8 µg/mg protein to 109.7 ± 16.9 µg/mg protein in 8 h incubations (Appendix A). The FA treatments likely pushed the limits of the maximum FA:protein ratio achievable in these cells as the total FA levels were rather similar in single and double supplementations (Table 2).

### 2.4. Cellular Triacylglycerol, Diacylglycerol and Ceramide Contents in Lipotoxicity

TGs, DGs and ceramides containing 16 and/or 18 carbon long saturated or monounsaturated fatty acyl chains were measured with HPLC-MS/MS.

According to our expectations, the applied FA supplementations triggered fat deposition in the hepatoma cells. Although palmitate alone caused a remarkable, nearly 4-fold increase in the overall TG level, the investigated cis or trans unsaturated FAs induced a much greater, more than 10-fold elevation in single supplementations (Figure 5 and Appendix A). The composition of accumulating fat was markedly different from the control. However, in the former case, the increment was largely due to the swelling of saturated species (TG 16:0_16:0_16:0, TG 16:0_16:0_18:0, TG 16:0_18:0_18:0 and TG 18:0_18:0_18:0), while in the latter, the vast majority of the deposited TGs were fully or dominantly unsaturated (TG 18:1_18:1_18:1, TG 16:0_18:1_18:1 and TG 18:0_18:1_18:1) (Appendix A).

When the cells were incubated with the combination of palmitate and any of the unsaturated FAs, the overall TG content grew higher than in the palmitate treated cells, and, at a 2:1 ratio at 8 h, grew as high or even higher than on the corresponding unsaturated FA. Importantly, the fat composition in these cells, i.e., the ratio of fully saturated types (TG 16:0_16:0_16:0, TG 16:0_16:0_18:0, TG 16:0_18:0_18:0 and TG 18:0_18:0_18:0) and those containing at least two unsaturated chains (TG 16:0_18:1_18:1, TG 18:0_18:1_18:1 and TG 18:1_18:1_18:1) was much more similar to the control (Figure 5 and Appendix A).

The moderate TG deposition was accompanied by an outstanding and progressive rise in the overall DG content in palmitate treated cells (Table 3, Figure 6 and Appendix A). In contrast to the 23-fold and 57-fold elevation caused by palmitate, only a 3-fold and less than 5-fold increase was detected upon oleate supplementation at 4 and 8 h, respectively. The two TFAs notably differed from oleate in this respect, as both increased the overall DG content more effectively, i.e., vaccenate induced a 4-fold and nearly 8-fold and elaidate caused a 5-fold and more than 9-fold elevation at 4 and 8 h, respectively.

It is remarkable that the vast majority, about 90% of DG buildup was due to the accumulation of 1,2-dipalmitoyl-glycerol (DG 16:0_16:0), while this fully saturated DG was barely measurable in control cells or in cells treated with any of the unsaturated FAs. Similarly, in the latter cells, fully unsaturated DG 18:1_18:1 contributed to about 80% of the total DGs (Figure 6 and Appendix A).

The inverse relationship between TG and DG levels was also evident in double supplementations. While the overall DG content increased above control in all the six combinations, it remained far below the value observed after single palmitate addition (Table 3). This reduction of DG accumulation was due mostly to the attenuation of 1,2-dipalmitoyl-glycerol buildup by the simultaneously added unsaturated FA (Figure 6 and Appendix A). The effect was dose-dependent as the 2:1 ratio caused a more pronounced difference than the 4:1 did in all cases; nevertheless, the TFAs were less efficient than oleate at both ratios at both incubation times. The major DG species in the control cells, palmitoyl-oleyl-glycerol (DG 16:0_18:1), showed a nearly 4-time elevation after palmitate treatment and no further increase upon the simultaneous addition of oleate; however, it grew substantially higher when a TFA, particularly vaccenate, was co-administered in the combination (Figure 6 and Appendix A).

Ceramides are produced during de novo synthesis or the salvage of sphingolipids, and both pathways utilize acyl-CoA molecules. Therefore, an acyl-CoA surplus in FA treated cells may cause alterations in the cellular ceramide content. The sharp contrast, which was seen between the modulation of TG and DG levels, was also observed in relation with TGs and ceramides (Table 4, Figure 7 and Appendix A). The total ceramide content of the cells reached the highest levels (765.3 ± 197.4 ng/mg protein at 4 h and 1 993.9 ± 100.5 ng/mg protein after 8 h) upon palmitate supplementation. These values represent a more than 4-fold and 13-fold increase compared to the controls (171.1 ± 12.3 and 149.3 ± 52.4 ng/mg protein, respectively). The accumulating ceramides contained saturated acyl groups, the concentration of palmitoyl-sphingosine (16:0) and stearoyl-sphingosine (18:0) increased parallelly while oleoyl-sphingosine remained near the limit of detection (Figure 7 and Appendix A).

When a single unsaturated FA was added to the cells, palmitoyl- and stearoyl-sphingosine were consistently unchanged, yet the alteration of unsaturated acyl-sphingosine (ceramide 18:1) depended greatly on the cis or trans nature of the FA. While no significant elevation was found in oleate treated cells, vaccenate induced a 25-fold (4 h) and 18-fold (8 h) elevation and elaidate caused an even higher, 39-fold (4 h) and 25-fold (8 h), increase compared to the control (Figure 7 and Appendix A).

All the three unsaturated FAs proved to be protective against palmitate-induced ceramide accumulation, similarly to what has been described regarding the DGs. Considering the 8 h-long co-supplementations, ceramide accumulation was approximately halved at the 4:1 combinations and quartered at the 2:1 molar ratios compared to palmitate alone (Table 4). Although the moderation of the total ceramide buildup was of nearly the same extent, and TFAs were only slightly less effective than oleate, a remarkable difference was again seen in the production of unsaturated acyl-sphingosine (ceramide 18:1), i.e., these ceramides only appeared in substantial amounts when TFAs were present. Intriguingly, their production seemed to be enhanced by palmitate because the half or quarter dose of a TFA co-administered with palmitate yielded the same or even more of these unusual ceramides as a large dose of the same TFA alone (Figure 7 and Appendix A).

### 2.5. Markers of Stress and Apoptosis in Fatty Acid Treated Cells

Stress-activated protein kinase, also referred to as c-Jun N-terminal kinase (SAPK/JNK), is involved in inflammatory signaling and is a suitable indicator for cellular stress of diverse origins. A massive phosphorylation of JNK was detected after palmitate treatment while the three unsaturated FAs had no such effect and P-JNK was hardly detectable by Western blot (Figure 8a). The combination treatments showed amelioration as JNK phosphorylation remained markedly below the palmitate-induced peak. Dose-dependence and a difference between cis and trans FAs could again be seen in this regard as the 2:1 ratio was more effective than the 4:1 in all three cases, and oleate almost completely abolished the sign of JNK phosphorylation (Figure 8a).

ER stress is a specific type of cellular stress, which is known to play a central role in lipotoxicity. ER dysfunction and the onset of UPR contribute significantly to the deleterious consequences of FA oversupply. This aspect was assessed in our experiments by detecting some of the key markers of early ER-stress at protein and mRNA levels. Eukaryotic initiation factor 2α (eIF2α) and inositol-requiring enzyme 1 (IRE1) stress receptor were both intensely phosphorylated upon palmitate supplementation while remaining unaffected by oleate, and the two TFAs had an intermediate effect, causing obvious but much milder signs of phosphorylation than seen in the palmitate-treated cells (Figure 8a). Palmitate-induced eIF2α and IRE1 phosphorylation was mitigated by the unsaturated FAs in a dose-dependent manner, and oleate was definitely more effective (Figure 8a).

IRE1 mediated splicing of X-box binding protein 1 (XBP1) mRNA is another widely used marker of the early ER-stress. The relative amount of spliced *XBP1* mRNA (*sXBP1*) was determined by using qPCR after 8 h-long FA treatments. The results correlated well with the changes of IRE1 phosphorylation revealed by using Western blot, as the highest *sXBP1* content was found after palmitate treatment, and none of the examined unsaturated FAs altered this parameter above the control level (Figure 8b). Additionally, oleate restrained palmitate-induced *XBP1* mRNA splicing the most efficiently, either at a 4:1 or at a 2:1 ratio in co-supplementations. The two TFAs also reduced the *sXBP1* content significantly compared to palmitate treatment, but their effectiveness was markedly lower, especially in the case of elaidate (Figure 8b).

The induction of the CCAAT-enhancer-binding protein homologous protein (CHOP) transcription factor is a key pro-apoptotic event when excessive ER-stress becomes destructive. Potential *CHOP* induction was assessed in our experiments at mRNA level by employing qPCR. A large increase was caused by palmitate while no change could be seen after unsaturated FA administration, either cis or trans (Figure 9a). Oleate helped the cells evade palmitate-induced *CHOP* induction in combination treatments, particularly at a 2:1 ratio. The two TFAs were also protective but to a significantly smaller extent than oleate (Figure 9a). To have a closer look into the activation of apoptosis, we examined the level of cleaved Caspase-3, the major effector caspase governing the process of apoptosis. In accordance with the observed stress signals, palmitate treatment resulted in the most pronounced activation of Caspase-3, while all unsaturated FAs were basically ineffective when applied on their own (Figure 9b). In combination treatments, oleate prevented the cleavage of Caspase-3 at the highest degree in a dose-dependent manner, and the TFAs exerted a similar, albeit milder, effect (Figure 9b).

### 2.6. Changes in Cellular Ultrastructure upon High Concentration Fatty Acid Supplementations

Morphological signs of ER damage, ER stress and intensified autophagy were sought by electron microscopy in HepG2 cells after 8 h-long treatment with FAs. The cytoplasm of control cells looked normal with mitochondria, numerous parallel rough ER cisternae, Golgi apparatus and a few lipid vesicles. There were very few old, dense autophagic vacuoles (Figure 10-Control).

While rough ER morphology did not change significantly in oleate-, elaidate- or vaccenate-treated cells, the ultrastructural signs of deranged ER morphology, i.e., large cracks alongside cisternae continued through the cytoplasm, were indicative of the presence of severe organelle stress in palmitate treated cells (Figure 10-panels of single treatments). Many small lipid vesicles with sharp, intact membrane boundaries were observed in oleate treated cells. Elaidate treatment caused small, rounded cell size filled with groups of small lipid vesicles with uneven boundary and slightly electrodense content. Vaccenate treated cells became large (~60 µm) and contained smaller and larger lipid vesicles in clusters. Some of the vesicles did not have a continuous membrane surface, fused with each other and enclosed slightly electrodense material and even some cytoplasmic remnants. Importantly, the similar appearance of the lipid vesicles can also be observed in the palmitate treated cells. Mitochondria did not show any defects upon fatty acid treatments (Figure 10).

Combined treatment had a diverse outcome. While oleate protected the cells from the cracks characteristic to palmitate treatment, elaidate and vaccenate had no visible effect (Figure 10—panels of double treatments). Cells after palmitate and oleate co-treatment showed significantly less and smaller cracks and aggregated large lipid vesicles with discontinuous membrane boundaries. Elaidate or vaccenate combined with palmitate produced a similarly harsh effect as palmitate separately. The cytoplasm was filled with large lipid vesicles with discontinuous membrane boundaries and dilated ER continuing in long, wavy cracks (Figure 10—panels of double treatments).

## 3. Discussion

FAs are building blocks of amphiphilic membrane lipids, such as phospholipids and sphingolipids, as well as of neutral storage and transport lipids, such as triacylglycerols and cholesteryl esters. They are also superb fuel molecules as their oxidation yields about twice as much energy as obtainable from carbohydrates or proteins. Food-derived FAs, as well as the endogenous FAs synthesized from the excess of carbohydrates and proteins in the fed state, are esterified and packed into lipoproteins for spreading via blood circulation, and they are mostly stored as depot fat in the adipocytes. It is in starvation and/or physical exercise that stored FAs are mobilized by TG hydrolysis in the adipose tissue and are made available for most of our cells as non-esterified or free FAs (FFAs). The FA composition of depot fat is largely species-specific as it is fundamentally determined by the substrate affinity of the acyltransferase enzymes, yet it is also modulated by the long-term dietary input of FAs [16]. The phrase ‘You are what you eat’ is not scientifically sound regarding the nucleic acids, proteins and carbohydrates but dietary FAs are admittedly integrated into human fat [17], plasma lipids [18], erythrocytes [19] and even in the skin epithelium [20]. While the FA profile of blood serum reflects short-term changes, that of the adipose tissue has been widely accepted as an indicator of the long-term dietary intake of FAs. Analysis of tissue specimens from 24 human subjects who died of heart disease showed a TFA content up to 12.2% in the adipose tissue, 14.4% in the liver, 9.3% in heart samples and 8.8% in aorta and the same in atheroma [21]. A good correlation has been represented between total dietary intake of TFAs and total TFA content of buttock adipose tissue [17], and, accordingly, TFAs have been also shown to contribute remarkably to the FFA fraction of human plasma lipids [18].

Adipocyte hypertrophy in obesity can lead to a local inflammation and insulin resistance in the strained adipose tissue. The consequently enhanced turnover of storage fat results in a sustained elevation of FFA levels in the blood plasma. The cells struggle to handle this non-physiological FFA oversupply; however, the induced damage, so-called lipotoxicity, greatly depends on what type of FA or FAs they are challenged with [22]. SFAs have long been known as the most deleterious FA species, and their major representative, palmitate, is the most widely used agent in cellular studies on lipotoxicity. Not only are the mono- or polyunsaturated FAs much more tolerable but they have also been found to ameliorate palmitate-induced cellular injuries when administered simultaneously in various models. The best studied monounsaturated FA, oleate, has been repeatedly shown to protect, among others, insulinoma cells [12,23] and hepatocytes or hepatocellular carcinoma cells [24,25] from palmitate-induced ER stress and apoptosis. These findings are in accordance with in vivo observations that vegetable oils, rich in unsaturated FAs, have more beneficial health effects than most animal fats of high SFA contents. Considering that TG synthesis offers an important means of unburdening the metabolism under the pressure of FFA abundance by channeling a substantial amount of acyl-CoA into lipid droplets, a plausible explanation for the phenomenon might be based on a limited capacity of the cells to produce all saturated TGs [22]. This assumption is further supported by the enhancement of SFA-induced toxicity when cellular FA desaturation is impeded [26].

Basically, all pathologies often proposed to be associated with TFA intake, i.e., cardiovascular diseases, non-alcoholic fatty liver disease (NAFLD), type 2 diabetes, are known to be related to obesity and thus to lipotoxicity, and the association between iTFA intake and obesity itself has been also reported [27]. This is why we thought it reasonable to test the representative iTFA, elaidate and rTFA, vaccenate in our established cellular model of lipotoxicity. As was mentioned in the Introduction, our previous findings showed that both TFAs acted almost exactly like oleate either alone or in combination with palmitate in insulinoma cells, yet a remarkable difference was also revealed regarding their incorporation into ceramides. These results led us to the conclusion that TFAs do indeed possess relevant metabolic specialties that might underlie their long term health effects. Although the diverse metabolic features of TFAs were not manifested in detectable differences in their short term toxicity in the insulinoma cells, we thought they were worth further investigating in vitro in cell cultures, so we extended our studies to HepG2 human hepatocellular carcinoma cells. In this cell line, we saw not only the same and even more metabolic differences but also evidence for a higher toxicity, or rather, a weaker protection by TFAs, and we have reason to believe that there is a causal relationship between the two phenomena.

Our current results correspond well with earlier data regarding the outstanding toxicity of palmitate as well as the relative tolerability and protective nature of oleate. The conditions, i.e., preparation, concentrations, ratios and incubation times were similar to those usually applied for these two FAs in HepG2 cells [24,25]. HepG2 cells were remarkably less sensitive to palmitate toxicity than the insulinoma cells were in our previous studies, and this might be at least partly due to the ability of hepatoma cells to grow lipid droplets. We saw clear signs of fat deposition in the electron micrographs of the FA-treated cells and, accordingly, measured high levels of various TG species in the corresponding cell samples. Palmitate treatment led to a nearly 4-fold increase in the overall TG content, while the less deleterious oleate at the same concentration caused an almost 12-fold elevation. It was evident that the cells were reluctant to accumulate dominantly-saturated TG molecules and much more ready to build up dominantly-unsaturated ones. When palmitate was co-administered with one fourth or half as much oleate, fat deposition intensified dramatically compared to that prompted by palmitate alone, and the composition and saturation level of deposited TGs became more like those seen in the untreated control cells. The cells incubated with a palmitate-oleate 2:1 combination stored as much TG as those treated with oleate alone but at a much more balanced and more control-like saturated:unsaturated ratio. The tendency of the cells to compile partly saturated and partly unsaturated TGs, if possible, is beautifully demonstrated by the near disappearance of all unsaturated trioleoyl-glycerol upon palmitate–oleate combinations. Importantly, the extent of TG accretion inversely correlated with that of DG and ceramide accumulation and also with the degree of stress, severity of cellular damages and intensity of cell death. Considering that DG and ceramides are potentially deleterious intermediates of lipid metabolism, our data further support the theory that unhindered TG synthesis is advantageous through draining the acyl-CoA pool toward less harmful and well storable compounds. It is also noteworthy that the relatively small amount of DGs emerging upon oleate treatment alone was almost entirely dioleoyl-glycerol, an otherwise minor fully unsaturated intermediate, which shows the ability of the cells to efficiently begin TG synthesis by inserting two unsaturated acyl groups, whereas only a minute amount of oleoyl-sphingosine could be detected in the same cells, clearly indicating that oleoyl-CoA is not a favorable substrate for ceramide synthases.

The two investigated TFAs acted similarly to oleate in most respects when they were applied on their own. They caused only a small reduction in cell viability even at high concentration and led to a remarkable deposition of TGs, including the dominantly unsaturated ones and to a relatively low buildup of DGs and ceramides, just like oleate. However, the appearance of a substantial amount of 18:1 ceramide species in TFA-treated cells emerges as a striking difference, especially in case of elaidate, and the ER stress markers (the phosphorylation of eIF2α and IRE1, the splicing of XBP1 mRNA and the induction of CHOP) also seemed to be slightly, yet not significantly, more elevated. As a matter of fact, these single treatments with an unsaturated FA at high (800 μM) concentration, although necessary parts of the study, do not reflect pathologically relevant conditions as faithfully as the combinational treatments when the unsaturated FA is added at lower (400 or 200 μM) concentrations together with the high dose palmitate. It is, therefore, the most important finding of our study that the stress, injuries and loss of viability were significantly more pronounced in the cells treated with palmitate and one of the TFAs simultaneously compared to those incubated with palmitate and oleate. Moreover, the correlations between the deleterious cellular effects and the concentrations of ceramides and DGs were also evident and hence were further established. Both TFAs counteracted the ceramide and the DG accumulating effect of palmitate in a dose-dependent manner but less effectively than oleate. The 18:1 ceramide levels upon the palmitate-TFA treatments rose above those seen in single TFA treatments despite the much lower TFA doses. This indicates that the incorporation of TFAs into ceramides reached enzyme saturation in our experimental conditions and also suggests that the abundance of palmitate, which leads to the enhanced palmitoylation of sphingosine and might also facilitate de novo sphinganine production [28], thus allowing the synthesis of extra elaidyl- and vaccenyl-ceramides.

HepG2 cells are widely used for investigating lipid metabolism and lipotoxicity. Despite the scarce cellular studies on TFAs, a serum-free version of this cell line has been treated with elaidate and/or vaccenate in earlier experiments. TFAs were added to HepG2-SF cells at 100 µM concentration in a 2:1 molar complex with human serum albumin repeatedly for 7 days to investigate their effect on cholesterol synthesis, esterification and export/import through modulating gene expression. In accordance with our data, both TFAs integrated efficiently into the lipidome, and they were not particularly toxic [29]. Nevertheless, after 7 days of supplementation, a larger amount of elaidate than vaccenate was measured in the cellular TGs, and only elaidate caused a considerable decrease in proliferation rate [29,30]. These differences were not observed in our experiments, which might be due to the different conditions, especially to the much shorter incubation times. The phenomenon detected after 7 days indicates a differential lipid metabolic response to the TFA isomers as a consequence of diverse alterations in gene expression, a factor that was overcome by direct metabolic effects in our short incubations. Accordingly, when rat hepatocytes were incubated with isotope-labeled FAs for 2 h, vaccenate and elaidate were shown to incorporate to the same extent in TGs, similarly to our findings, and only phospholipids acquired more elaidate than vaccenate [31]. A more intense incorporation of elaidate into phospholipids in rat hepatocytes was also observed in another study [32], and a similar difference was repeatedly found in human umbilical vein endothelial cells (HUVEC) [33,34]. However, no significant difference was found between the overall incorporation of oleate, vaccenate or elaidate in another endothelial cell line (EA.hy926) when incubated for 48 h at low concentration (1 to 50 μM) of these FAs [35]. It is interesting and somewhat contrary to the above observations that the TG content of secreted lipoproteins was slightly more elevated with vaccenate than with elaidate [31]. It is in accordance with our findings that both cis and trans C-18 unsaturated FAs caused a remarkably more pronounced TG accumulation in human intestinal Caco-2 cells, compared to C-16 and C-18 saturated ones [36], and, accordingly, both oleate and elaidate triggered much more lipid droplet deposition than palmitate in U937 human monocytes [37].

In contrast to the plethora of studies elucidating the extent and molecular mechanisms of palmitate-induced cellular toxicity and the mitigating effect of oleate, very few in vitro experiments have been performed to reveal the toxicity or the potential protectiveness of TFAs, and the detailed pathogenic mechanisms and specific molecular targets underlying TFA-related disorders are largely unknown. A recent review that thoroughly summarizes the available data regarding the mechanism of action of TFAs [38] refers to many relevant in vivo studies and only a few cellular research, most of which focused on the modulation of inflammatory signaling and oxidative stress. The role of ER stress and JNK-mediated apoptosis in lipotoxicity in various cell types, including hepatocytes, is well established [38]. The ER stress-inducing effect of the iTFA elaidate has been demonstrated in SH-SY5Y neuroblastoma cells. Elaidate induced the ER chaperone glucose-regulated protein 78 (GRP78), and transiently upregulated activating transcription factor 4 (ATF4) and CHOP in this model [39]. However, TFAs did not induce ER stress or autophagy in human osteosarcoma U2OS cells. Moreover, several TFAs including elaidate and vaccenate efficiently suppressed ER stress or autophagy induced by palmitate [40].

Although some information is available on the incorporation of TFAs into TGs and phospholipids in various cell types, very little is known about their cellular metabolism, particularly about their effect on ceramide and diacylglycerol accumulation. The potential role of these TFA-containing lipid intermediates in pathological conditions is undeservedly neglected. Metabolism-related toxicity of TFAs was compared to that of oleate in HUVEC cells, and they showed a much greater sensitivity to TFAs than HepG2 cells. Both vaccenate and elaidate, especially the latter, were found to be more toxic than oleate even at 100 μM concentration, and this was related to the level of their incorporation into membrane phospholipids [33]. The potential role of ceramides or DGs was not investigated in this study either.

In summary, uptake of TFAs and their incorporation into TGs and membrane phospholipids have been demonstrated in various cells, but their effects on ceramide and DG accumulation have not been investigated, nor have they been correlated with ER stress, JNK activation and apoptosis in single treatments and in combination with palmitate. We believe that our results provide convincing evidence for the existence of characteristic differences between the utilization of cis and trans FAs in lipid synthesis. Considering the pathological role of ceramides [15], it can be reasonably assumed that these differences play a causal role in the in vivo-observed health effects of the latter. We also hope that the presented phenomena will attract more attention to this area because it is definitely worth further investigation.

## 4. Materials and Methods

### 4.1. Materials Used

Culture medium and supplements were purchased from Life Technologies. Palmitate, oleate, elaidate, vaccenate, FA free bovine serum albumin, trans-vaccenic acid methyl ester, methyl oleate, methyl palmitate, methyl palmitoleate methyl stearate, 1,3-diheptadecanoyl-glycerol (d5) (>99%), 1,2,3-triheptadecanoyl-glycerol (>99%), C17:0 ceramide (>99%), n-hexane, methanol (gradient grade) and isopropanol (gradient grade) and boron trifluoride–methanol solution were purchased from Merck. All other chemicals used in this study were of analytical grade. All experiments and measurements were carried out by using Millipore ultrapure water.

### 4.2. Cell Culture

HepG2 hepatocellular carcinoma cell line was purchased from Sigma and cultured in DMEM medium, containing 4.5 g/l L-glutamine, 4.5 g/l D-glucose, free of pyruvate and supplemented with 10% fetal bovine serum and 1% antibiotics (Thermo Scientific, Waltham, MA, USA) at 37 °C in humidified atmosphere containing 5% CO_2_.

### 4.3. Cell Treatment with BSA-Conjugated Fatty Acids

Palmitate, elaidate, oleate and vaccenate (Sigma, Burlington, MA, USA) were diluted in ethanol (Molar Chemicals, Budapest, Hungary) to a concentration of 100 mM, conjugated with 20% FA free BSA (Sigma, Burlington, MA, USA) in a 1:4 ratio at 50 °C for 1 h. The working solution for FA treatments was always freshly prepared in FBS-free and antibiotic-free medium in a 0.2, 0.4 or 0.8 mM final concentration. The culture medium was replaced by FBS-free and antibiotic-free medium at 1 h before the cells were treated with FAs for 8–24 h at 70–80% confluence in 6-well plates (for Western blot; qPCR; electron microscopy and analysis of ceramides, DGs, TGs and fatty acid profile) or in 96-well plates (for cell viability assay and the detection of apoptosis and necrosis).

### 4.4. Analysis of Lipid Contents

Cell media were withdrawn for FA analysis and stored at −20 °C until use. For the analysis of cell-associated FAs, ceramides, DGs and TGs, cells were washed once with PBS, then harvested in 100 µL PBS by scraping. The samples were then sedimented in a benchtop centrifuge (5 min, 1500 rpm, 24 °C), and the supernatants were discarded. The cells were suspended in PBS, and the protein concentration of the cell suspension was measured, as mentioned, in Western blot analysis. 50 µL of each suspension was transferred to a clear crimp vial for GC-FID measurement of fatty acids [41], and 50 µL was transferred to a micro-centrifuge tube for the HPLC-MS/MS analysis of TGs, DGs and ceramides [42].

#### 4.4.1. GC-FID Analysis of Fatty Acid Profiles

An amount of 100 µL methanol containing 2 W/V% NaOH was added to 200 µL medium sample, and 150 µL of methanol containing 2 W/V% NaOH was added to the 50 µL cell suspension in the crimp vials. The samples were incubated at 90 °C for 30 min and then cooled to room temperature. An amount of 400 µL of methanol containing 13–15% of boron trifluoride was added to the samples, and the vials were incubated at 90 °C for 30 min. After cooling to room temperature, 200 µL of saturated NaCl solution and 500 µL of n-hexane were added. Fatty acid methyl esters were extracted to the upper phase containing n-hexane. An amount of 400 µL of this phase was dried, reconstituted in 200 µL n-hexane and transferred to a vial for GC analysis.

Samples of 1 µL volume were separated in a Zebron ZB-88 capillary column (60 m × 0.25 mm i.d., 0.20 µm film thickness) by using a Shimadzu GC-2014 gas chromatograph equipped with a Shimadzu AOC-20s autosampler and a flame ionization detector (FID). The carrier gas was hydrogen at 35 cm/sec velocity. The injector and detector temperature was 250 °C and the oven temperature was ramped from 100 °C to 210 °C at a rate of 4 °C/min. Quantitative analysis was based on a five-point calibration in the concentration range of 1–200 μg/mL for each FA [41].

#### 4.4.2. HPLC-MS/MS Analysis of Triacylglycerols, Diacylglycerols and Ceramides

Cells were pelleted by centrifugation (5 min, 1500 rpm, 24 °C) and resuspended in a mixture of methanol–isopropanol (1:1 ratio) containing ceramide 17:0 (500 ng/mL), DG 17:0_17:0 d5 (1 µg/mL) and TG 17:0_17:0_17:0 (1 µg/mL) internal standards. The samples were homogenized with an ultrasonic sonotrode and centrifuged (10 min, 13,400 rpm, 24 °C). The supernatants were transferred to vials for HPLC-MS/MS analysis.

5 µL samples were injected in the HPLC (Agilent 1100). A Kinetex^®^ 5 µm, C8 100 Å, LC (100 × 3 mm) column was used with a gradient elution of 10 mM ammonium-acetate (mobile phase A) and methanol (mobile phase B) and isopropanol (mobile phase C): 0 min at 10% A, 90% B, 0% C; 5 to 7 min at 5% A, 95% B, 0% C; 14 to 16 min at 5% A, 55% B and 40% C; 17 to 23 min at 10% A, 90% B, 0% C. Representative chromatogram of a cell sample is shown in Appendix A. Ceramide, diacylglycerol and triacylglycerol species were detected using a triple quadrupole mass spectrometer (SCIEX 3500). The instrument was used in positive multiple reaction monitoring mode. The m/z values of precursor and product ions for each analyte are shown in Appendix A. The ion spray temperature was set to 450 °C and the voltage to 5500 V. Quantitative analysis was based on internal standard method by using non-physiological metabolite analogues, i.e., ceramide 17:0 (500 ng/mL), DG 17:0_17:0 d5 (1 µg/mL) and TG 17:0_17:0_17:0 (1 µg/ml) internal standards.

### 4.5. Electron Microscopy

For electron microscopy, cells were grown on PDL coated glass cover slips for 24 h and treated at 70–80% confluence, as described above. The medium was removed from the cell cultures, the cells were washed twice with PBS and fixed in 3.2% paraformaldehyde, 0.5% glutaraldehyde, 1% sucrose and 0.028% CaCl_2_ in 0.1 M sodium cacodylate, pH 7.4, overnight at 4 °C, and then washed and embedded in Durcupan (Fluka, Darmstadt, Germany). Ultrathin sections were contrasted with uranyl acetate and with Reynold’s lead citrate and examined and photographed using a Jeol JEM-1011 electron microscope, operating at 80 kV, equipped with Olympus Morada CCD camera using Olympus iTEM 5.1 (TEM imaging platform) software.

### 4.6. Cell Viability

Cell viability was assessed by using the Colorimetric (MTT) Kit for Cell Survival and Proliferation (Millipore, Burlington, MA, USA) according to the manufacturer’s instructions. MTT-derived formazan was measured at 530 nm test and 630 nm reference wavelengths in a multiscan spectrophotometer (Thermo Scientific, Waltham, MA, USA). Cell viability was expressed as the percentage of viable cells in the total cell population.

### 4.7. Western Blot Analysis

Cells were washed twice with PBS and harvested in 100 µL lysis buffer by scraping. The lysis buffer contained 0.1% SDS, 5 mM EDTA, 150 mM NaCl, 50 mM Tris, 1% Tween 20, 1 mM Na_3_VO_4_, 1 mM PMSF, 10 mM benzamidine, 20 mM NaF, 1 mM pNPP and protease inhibitor cocktail. The lysates were centrifuged in a benchtop centrifuge (10 min, 10,000 rpm, 4 °C). The protein concentration of the supernatant was measured using Pierce BCA Protein Kit Assay (Thermo Scientific, Waltham, MA, USA) and the samples were stored at −20 °C until use.

Samples (20 μg protein) were electrophoresed in 10–12–15% SDS polyacrylamide gels and transferred to PVDF membranes (Millipore). Primary and secondary antibodies were applied overnight at 4 °C and for 1 h at room temperature, respectively. Equal protein loading was validated by detection of glyceraldehyde 3-phosphate dehydrogenase (GAPDH), with a mouse monoclonal anti-GAPDH (Santa Cruz, Dallas, TX, USA, sc-32233) antibody, at 1:20,000 dilution, as a constitutively expressed reference protein. Primary antibodies: rabbit anti-phospho-IRE1 (S724) (#ab48187) from Abcam, rabbit anti-Cleaved Caspase-3 (#9661), rabbit anti-phospho-SAPK/JNK (THR183/Tyr185) (#9251S), rabbit anti-SAPK/JNK (#9252S), rabbit anti-phospho-eIF2α (#9721) and rabbit anti-eIF2α (#9722) from cell signaling. Secondary antibodies: horseradish peroxidase (HRP)-conjugated goat anti-rabbit IgG-HRP (#7074) and HRP-conjugated horse anti-mouse IgG-HRP (#7076) from cell signaling. HRP was detected with chemiluminescence using SuperSignal West Pico Chemiluminescent Substrate (Thermo Scientific).

### 4.8. qPCR Analysis

Total RNA was purified from the cells by using RNeasy Plus Mini Kit (Qiagen, Düsseldorf, Germany) following the manufacturer’s instruction. cDNA was produced by reverse transcription of 0.5 μg DNA-free RNA samples using the SuperScript III First-Strand Synthesis System for RT-PCR Kit (Invitrogen, Waltham, MA, USA). Quantitative qPCR assay was performed in 20 µL final volume containing 5 µL cDNA diluted to one twentieth, 1 × PowerUp^TM^SYBR^TM^Green Master Mix, 0.5 µM target sequence specific forward and reverse primers using Applied Biosystems 7300 Real-Time PCR System (Thermo Scientific, Waltham, MA, USA). Spliced *XBP1*, total *XBP1* and *CHOP* sequences were amplified by 5′– CTG AGT CCG AAT CAG GTG CAG –3′, 5′–ATC CAT GGG GAG ATG TTC TGG –3′, 5′– TGG CCG GGT CTG CTG AGT CCG –3′, 5′–ATC CAT GGG GAG ATG TTC TGG –3′ and 5′– GTA CCT ATG TTT CAC CTC CTG G –3′, 5′– TGG AAT CTG GAG AGT GAG GG –3′ primer pairs, respectively. *GAPDH* cDNA was also amplified as a reference control using 5′– GTC CAC TGG CGT CTT CAC CA –3′ and 5′– GTG GCA GTG ATG GCA TGG AC –3′ primers. Denaturation at 95 °C, 2 min was followed by 40 cycles (95 °C, 15 s; 55 °C, 15 s and 72 °C 1 min). Reactions were performed using RNase-free water as negative control and CT-values were set in the exponential range of the amplification. Relative expression levels were determined as 2^−ΔΔCT^, where ΔΔC_T_ values correspond to the difference between the C_T_-values of the target and the internal control genes.

### 4.9. Statistics

Data are presented in the diagrams as mean values ± S.D. and were compared by ANOVA with Tukey’s multiple comparison post hoc test using GraphPad Prism 6 software. Differences of a P value below 0.05 were considered to be statistically significant.

## 5. Conclusions

This study provides important in vitro evidence for the different metabolism of cis and trans C18:1 FAs in correlation with their different toxicity in human cells. Compared to oleate, the two representative TFAs, elaidate and vaccenate, induced slightly more pronounced stress and cell damage on their own, and they were significantly less protective against palmitate toxicity in our experiments. The most remarkable difference that we found between the metabolism of cis and trans C18:1 FAs is the more pronounced incorporation of the latter in the potentially harmful lipid intermediates ceramides and diacylglycerols. Our findings further support the role of ceramide and DG accumulation in the development of cellular dysfunctions in metabolic diseases and offer a potential explanation to the long-term health effects, which are attributed to trans fat on the basis of in vivo data.

## Figures and Tables

**Figure 1 ijms-23-07298-f001:**
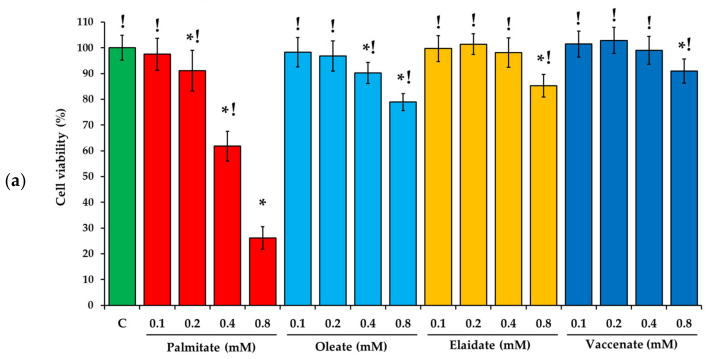
Viability of cell treated with fatty acids at various concentrations. Cells were treated with (**a**) BSA-conjugated palmitate, oleate, elaidate or vaccenate (100–800 µM) or with (**b**) palmitate (800 µM) and oleate, elaidate or vaccenate at a 4:1 or 2:1 molar ratio (200 or 400 µM) at 70–80% confluence for 24 h. Cell viability was assessed by the Colorimetric (MTT) Kit for Cell Survival and Proliferation (Millipore) and expressed as the percentage of the control. Data are shown as mean values ± S.D.; *n* = 18; statistically significant differences: * *p* < 0.05 vs. BSA treated control; ^!^
*p* < 0.05 vs. 800 µM palmitate treated samples; ^#^
*p* < 0.05 vs. palmitate and oleate treated cells in case of combinational treatments at the corresponding ratio.

**Figure 2 ijms-23-07298-f002:**
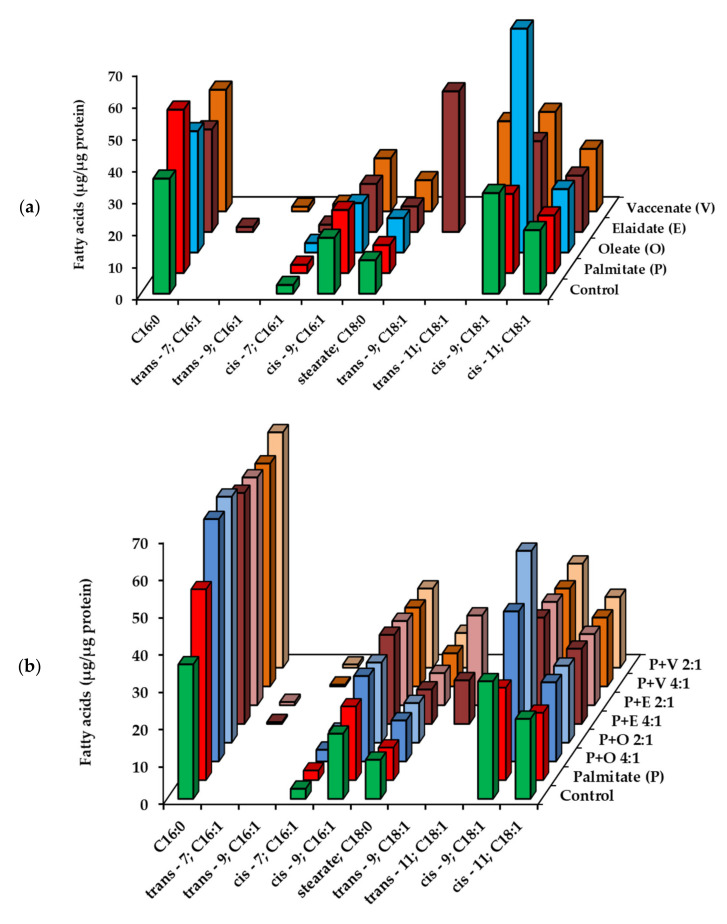
Fatty acid profile of HepG2 cells treated with fatty acids at sub-toxic concentrations. Cells were treated with (**a**) BSA-conjugated palmitate, oleate, elaidate or vaccenate (50 µM) or with (**b**) palmitate (50 µM) and oleate, elaidate or vaccenate at a 4:1 or 2:1 molar ratio (12.5 or 25 µM) at 70–80% confluence for 8 h. The amount of relevant fatty acids was measured in the cell samples by GC-FID after saponification and methylation. Data were normalized to the total protein content of the samples and are shown as mean values of at least four independent experiments (see the mean and S.D. values in Appendix A).

**Figure 3 ijms-23-07298-f003:**
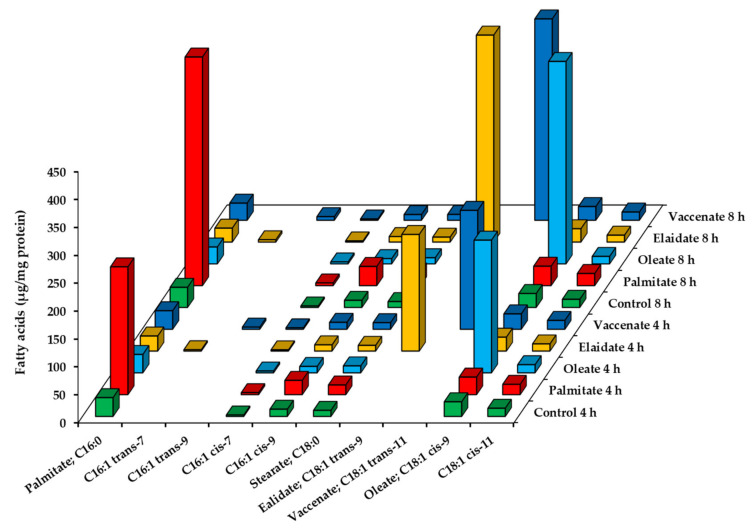
Fatty acid profile of HepG2 cells supplemented with a single fatty acid at high concentration. Cells were treated with BSA-conjugated palmitate, oleate, elaidate or vaccenate (800 µM) at 70–80% confluence for 4 h or 8 h. The amount of relevant fatty acids was measured by GC-FID after saponification and methylation. Data were normalized to the total protein content of the samples and are shown as mean values of at least four independent experiments (see the mean and S.D. values in Appendix A).

**Figure 4 ijms-23-07298-f004:**
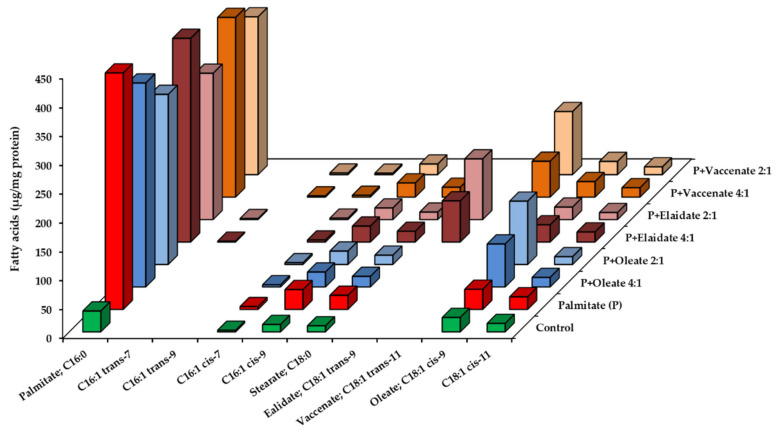
Fatty acid profile of HepG2 cells supplemented with high concentration palmitate and a monounsaturated fatty acid. Cells were treated with BSA-conjugated palmitate (800 µM) and oleate, elaidate or vaccenate at a 4:1 or 2:1 molar ratio (200 or 400 µM) at 70–80% confluence for 8 h. The amount of relevant fatty acids was measured by GC-FID after saponification and methylation. Data were normalized to the total protein content of the samples and are shown as mean values of at least four independent experiments (see the mean and S.D. values in Appendix A).

**Figure 5 ijms-23-07298-f005:**
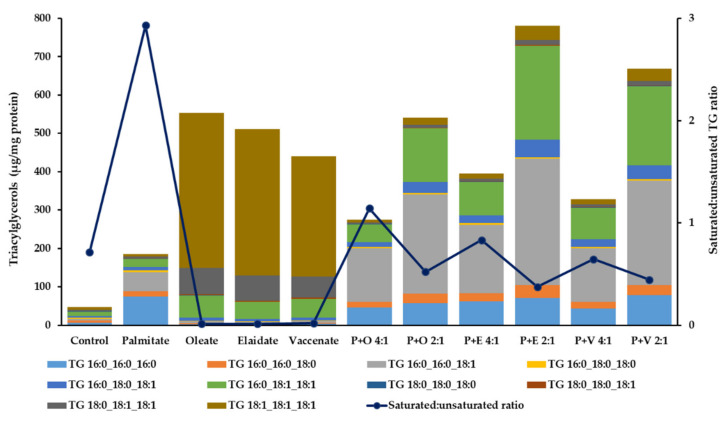
Triglyceride content of hepatoma cells in lipotoxicity. Cells were treated with BSA-conjugated palmitate, oleate, elaidate or vaccenate (800 µM) or with palmitate (800 µM) and oleate, elaidate or vaccenate at 4:1 or 2:1 molar ratio (200 or 400 µM) at 70–80% confluence for 8 h. The amount of relevant triglycerides (columns) was measured by LC-MS/MS. Data were normalized to the total protein content of the samples and are shown as the mean values of four independent experiments (see the mean and S.D. values in Appendix A). The ratio of the mean values for fully saturated types (TG 16:0_16:0_16:0, TG 16:0_16:0_18:0, TG 16:0_18:0_18:0 and TG 18:0_18:0_18:0) and those containing at least two unsaturated chains (TG 16:0_18:1_18:1, TG 18:0_18:1_18:1 and TG 18:1_18:1_18:1) was calculated for each treatment (full circles connected with a continuous line for better visibility, right axis).

**Figure 6 ijms-23-07298-f006:**
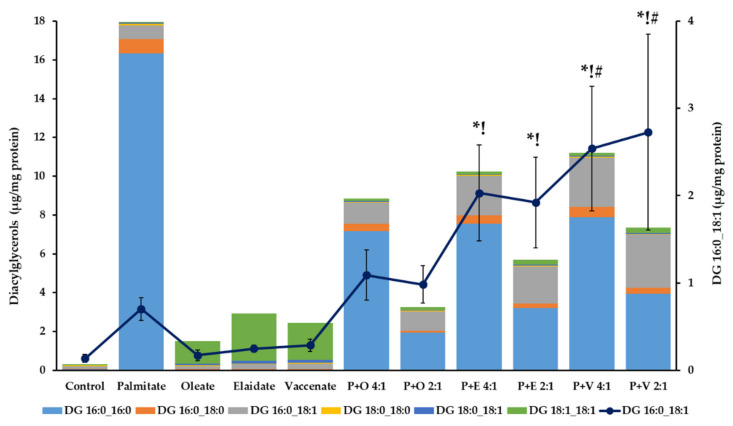
Diglyceride content of hepatoma cells in lipotoxicity. Cells were treated with BSA-conjugated palmitate, oleate, elaidate or vaccenate (800 µM) or with palmitate (800 µM) and oleate, elaidate or vaccenate at a 4:1 or 2:1 molar ratio (200 or 400 µM) at 70–80% confluence for 8 h. The amount of six relevant diglycerides (columns) was measured by LC-MS/MS. Data were normalized to the total protein content of the samples and are shown as mean values of at least four independent experiments (see the mean and S.D. values in Appendix A). DG 16:0_18:1 contents are also shown separately (line, right axis) as mean values ± S.D.; *n* ≥ 4; statistically significant differences: * *p* < 0.05 vs. BSA treated control; ^!^
*p* < 0.05 vs. palmitate treated samples; ^#^
*p* < 0.05 vs. palmitate and oleate treated cells in case of combinational treatments at the corresponding ratio.

**Figure 7 ijms-23-07298-f007:**
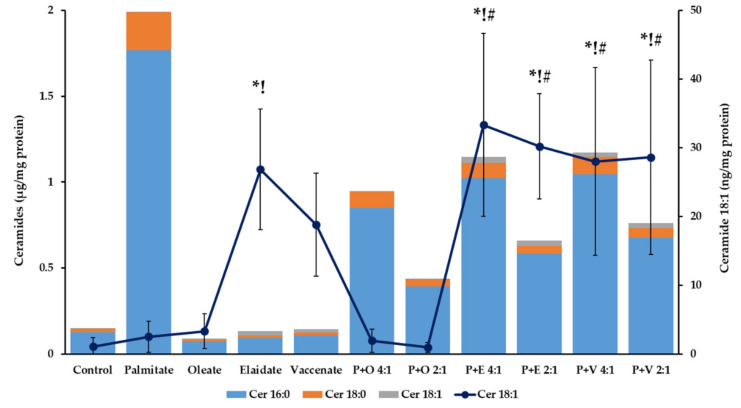
Ceramide content of hepatoma cells in lipotoxicity. Cells were treated with BSA-conjugated palmitate, oleate, elaidate or vaccenate (800 µM) or with palmitate (800 µM) and oleate, elaidate or vaccenate at a 4:1 or 2:1 molar ratio (200 or 400 µM) at 70–80% confluence for 8 h. The amount of three relevant ceramides was measured by LC-MS/MS. Data were normalized to the total protein content of the samples and are shown as mean values of at least four independent experiments (see the mean and S.D. values in Appendix A). Ceramide 18:1 contents are also shown separately (line, right axis) as mean values ± S.D.; *n* ≥ 3; statistically significant differences: * *p* < 0.05 vs. BSA treated control; ^!^
*p* < 0.05 vs. palmitate treated samples; ^#^
*p* < 0.05 vs. palmitate and oleate treated cells in case of combinational treatments at the corresponding ratio.

**Figure 8 ijms-23-07298-f008:**
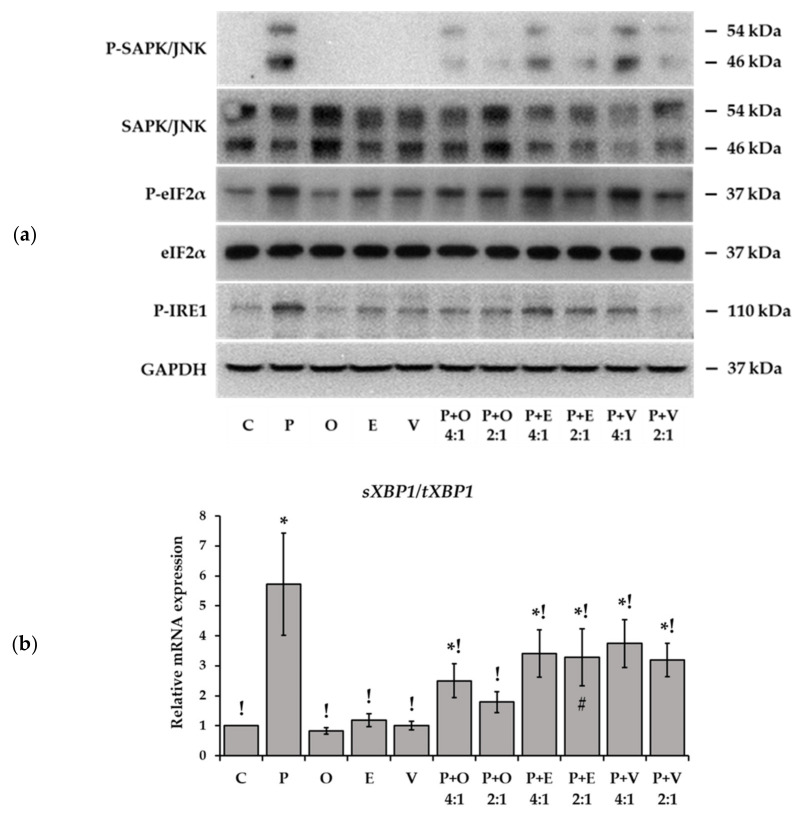
Protein and mRNA markers of ER stress. Cells were treated with BSA-conjugated palmitate, oleate, elaidate or vaccenate (800 µM) or with palmitate (800 µM) and oleate, elaidate or vaccenate at a 4:1 or 2:1 molar ratio (200 or 400 µM) at 70–80% confluence for 8 h. (**a**) Phosphorylated and total SAPK/JNK isoforms (P-JNK 1/2), phosphorylated and total eIF2α (P-eIF2α) and phosphorylated IRE1 proteins were detected by Western blot in the cell lysates. GAPDH was used as a constitutive reference protein. The images show typical results of two independent experiments with two parallels. (**b**) cDNA was prepared and spliced *sXBP1* and total *tXBP1* were detected by qPCR. Relative expression levels were determined as *sXBP1/tXBP1* ratios. Data are shown as mean values ± S.D.; *n* = 6; statistically significant differences: * *p* < 0.05 vs. BSA treated control; ^!^
*p* < 0.05 vs. palmitate treated samples; ^#^
*p* < 0.05 vs. palmitate and oleate treated cells in case of combinational treatments at the corresponding ratio.

**Figure 9 ijms-23-07298-f009:**
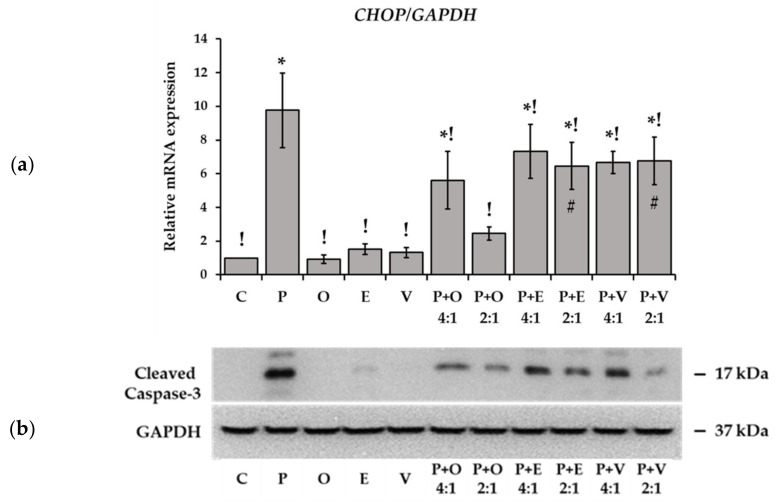
Protein and mRNA markers of ER stress related apoptosis. Cells were treated with BSA-conjugated palmitate, oleate, elaidate or vaccenate (800 µM) or with palmitate (800 µM) and oleate, elaidate or vaccenate at a 4:1 or 2:1 molar ratio (200 or 400 µM) at 70–80% confluence for 8 h. (**a**) cDNA was prepared and *CHOP* expression was detected by qPCR. *GAPDH* was used as a constitutive reference gene. Relative expression levels were determined as *CHOP/GAPDH* ratios. Data are shown as mean values ± S.D.; *n* = 6; statistically significant differences: * *p* < 0.05 vs. BSA treated control; ^!^
*p* < 0.05 vs. palmitate treated samples; ^#^
*p* < 0.05 vs. palmitate and oleate treated cells in case of combinational treatments at the corresponding ratio. (**b**) Cleaved caspase-3 was detected in cell lysates by Western blot. Glyceraldehyde 3-phosphate dehydrogenase (GAPDH) was used as a constitutive reference protein. The image shows a typical picture of two independent experiments with two parallels.

**Figure 10 ijms-23-07298-f010:**
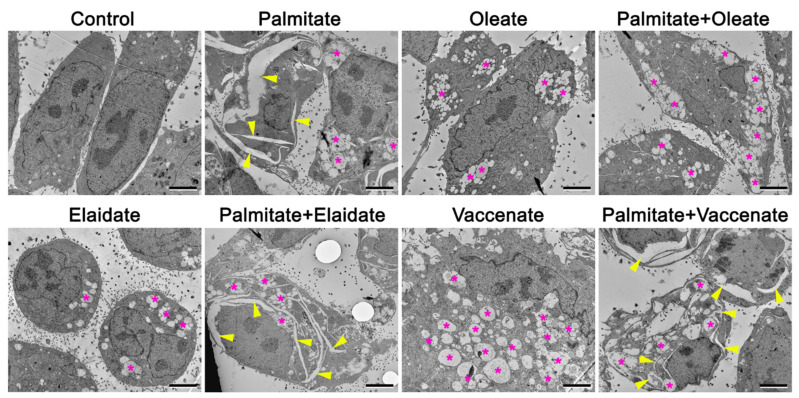
Electron micrographs of fatty acid treated cells. Cells were treated with BSA-conjugated palmitate, oleate, elaidate or vaccenate (800 µM) or with palmitate (800 µM) and oleate, elaidate or vaccenate at a 2:1 molar ratio (400 µM) at 70–80% confluence for 8 h. Yellow arrowheads point to cracks along ER cisterns, magenta asterisks label groups of lipid vesicles. Scale bar 2 µm.

**Table 1 ijms-23-07298-t001:** Overall triglyceride, diglyceride and ceramide contents of the cells treated with fatty acids at sub-toxic concentrations. Cells were treated with BSA-conjugated palmitate, oleate, elaidate or vaccenate (50 µM) or with palmitate (50 µM) and oleate, elaidate or vaccenate at 4:1 or 2:1 molar ratio (12.5 or 25 µM) at 70–80% confluence for 8 h. The overall amount of ten triglycerides, six diglycerides and three ceramides was measured by LC-MS/MS. Data were normalized to the total protein content of the samples and are shown as mean values ± S.D.; *n* ≥ 4; statistically significant differences: * *p* < 0.05 vs. BSA treated control; ^!^
*p* < 0.05 vs. palmitate treated samples.

Treatment (8 h)	TGs ^1^ (μg/mg Protein)	DGs ^2^ (ng/mg Protein)	Ceramides ^3^ (ng/mg Protein)
Control	31.4 ± 11.4 ^!^	333.9 ± 39.1 ^!^	159.6 ± 20.2 ^!^
Palmitate	64.4 ± 24.0 *	560.2 ± 100.3 *	237.3 ± 12.3 *
cis-Oleate	52.7 ± 17.2	388.9 ± 81.9 ^!^	98.3 ± 16.4 *^,!^
trans-Elaidate	45.8 ± 13.6	443.0 ± 68.0 ^!^	91.9 ± 16.4 *^,!^
trans-Vaccenate	48.4 ± 8.6	365.1 ± 67.1 ^!^	120.7 ± 25.4 *^,!^
Palmitate + Oleate (4:1)	59.0 ± 15.3	506.4 ± 72.0 *	208.4 ± 37.3 *
Palmitate + Oleate (2:1)	97.7 ± 37.7 *^,!^	514.4 ± 94.7 *	210.6 ± 29.5 *
Palmitate + Elaidate (4:1)	56.1 ± 16.9	514.4 ± 45.0 *	202.5 ± 35.6 *
Palmitate + Elaidate (2:1)	96.3 ± 32.0 *^,!^	586.1 ± 114.8 *	207.5 ± 33.5 *
Palmitate + Vaccenate (4:1)	61.1 ± 12.9 *	538.8 ± 113.7 *	214.9 ± 38.4 *
Palmitate + Vaccenate (2:1)	98.2 ± 29.4 *^,!^	502.9 ± 107.1 *	212.4 ± 32.9 *

^1^ Sum of the amounts of ten triglycerides: TG 16:0_16:0_16:0; TG 16:0_16:0_18:0; TG 16:0_16:0_18:1; TG 16:0_18:0_18:0; TG 16:0_18:0_18:1; TG 16:0_18:1_18:1; TG 18:0_18:0_18:0; TG 18:0_18:0_18:1; TG 18:0_18:1_18:1; and TG 18:1_18:1_18:1 (see also Appendix A). ^2^ Sum of the amounts of six diglycerides: DG 16:0_16:0; DG 16:0_18:0; DG 16:0_18:1; DG 18:0_18:0; DG 18:0_18:1 and DG 18:1_18:1 (see also Appendix A). ^3^ Sum of the amounts of three ceramides: Cer 16:0; Cer 18:0 and Cer 18:1 (see also Appendix A).

**Table 2 ijms-23-07298-t002:** Overall cell-associated fatty acids in lipotoxicity. Cells were treated with BSA-conjugated palmitate, oleate, elaidate or vaccenate (800 µM) or with palmitate (800 µM) and oleate, elaidate or vaccenate at a 4:1 or 2:1 molar ratio (200 or 400 µM) at 70–80% confluence for 4 or 8 h. The overall amount of 10 relevant fatty acids was measured by GC-FID after saponification and methylation. Data were normalized to the total protein content of the samples and are shown as mean values ± S.D.; *n* ≥ 4; statistically significant differences: * *p* < 0.05 vs. BSA treated control; ^!^
*p* < 0.05 vs. palmitate treated samples.

	Overall FA Content ^1^ (μg/mg Protein)	
Treatment	at 4 h	at 8 h
Control	101.0 ± 21.1 ^!^	101.5 ± 30.7 ^!^
Palmitate	325.3 ± 43.8 *	507.0 ± 101.6 *
cis-Oleate	312.4 ± 36.3 *	429.9 ± 47.2 *
trans-Elaidate	299.2 ± 78.6 *	456.7 ± 114.6 *
trans-Vaccenate	320.0 ± 46.7 *	460.3 ± 80.1 *
Palmitate + Oleate (4:1)	326.5 ± 58.0 *	492.0 ± 100.5 *
Palmitate + Oleate (2:1)	312.1 ± 25.3 *	460.2 ± 97.8 *
Palmitate + Elaidate (4:1)	336.8 ± 56.5 *	522.3 ± 120.9 *
Palmitate + Elaidate (2:1)	331.0 ± 34.2 *	429.4 ± 93.4 *
Palmitate + Vaccenate (4:1)	323.7 ± 70.8 *	463.0 ± 83.2 *
Palmitate + Vaccenate (2:1)	348.6 ± 67.1 *	457.1 ± 77.6 *

^1^ Sum of the amounts of ten fatty acids: C16:0; C16:1 trans-7; C16:1 trans-9; C16:1 cis-7; C16:1 cis-9; C18:0; C18:1 trans-9; C18:1 trans-11; C18:1 cis-9 and C18:1 cis-11 (see also Appendix A).

**Table 3 ijms-23-07298-t003:** Overall diglyceride content of hepatoma cells in lipotoxicity. Cells were treated with BSA-conjugated palmitate, oleate, elaidate or vaccenate (800 µM) or with palmitate (800 µM) and oleate, elaidate or vaccenate at a 4:1 or 2:1 molar ratio (200 or 400 µM) at 70–80% confluence for 4 or 8 h. The overall amount of six relevant diglycerides was measured by LC-MS/MS. Data were normalized to the total protein content of the samples and are shown as mean values ± S.D.; *n* ≥ 4; statistically significant differences: * *p* < 0.05 vs. BSA treated control; ^!^
*p* < 0.05 vs. palmitate treated samples.

	Overall DG Content ^1^ (μg/mg Protein)	
Treatment	at 4 h	at 8 h
Control	0.3 ± 0.1 ^!^	0.3 ± 0.1 ^!^
Palmitate	8.1 ± 2.3 *	18.0 ± 2.3 *
cis-Oleate	1.1 ± 0.2 ^!^	1.5 ± 0.4 ^!^
trans-Elaidate	1.7 ± 0.3 ^!^	2.9 ± 0.4 ^!^
trans-Vaccenate	1.4 ± 0.3 ^!^	2.4 ± 0.4 ^!^
Palmitate + Oleate (4:1)	3.7 ± 1.1 *^,!^	8.9 ± 3.0 *^,!^
Palmitate + Oleate (2:1)	2.1 ± 0.6 ^!^	3.3 ± 1.1 ^!^
Palmitate + Elaidate (4:1)	4.3 ± 1.6 *^,!^	10.3 ± 3.1 *^,!^
Palmitate + Elaidate (2:1)	2.8 ± 0.9 ^!^	5.7 ± 1.6 ^!^
Palmitate + Vaccenate (4:1)	5.1 ± 1.8 *^,!^	11.2 ± 2.7 *
Palmitate + Vaccenate (2:1)	3.8 ± 1.4 *^,!^	7.3 ± 2.9 *^,!^

^1^ Sum of the amounts of six types of diglycerides: DG 16:0_16:0, DG 16:0_18:0, DG 16:0_18:1, DG 18:0_18:0, DG 18:0_18:1 and DG 18:1_18:1.

**Table 4 ijms-23-07298-t004:** Overall ceramide content of hepatoma cells in lipotoxicity. Cells were treated with BSA-conjugated palmitate, oleate, elaidate or vaccenate (800 µM) or with palmitate (800 µM) and oleate, elaidate or vaccenate at a 4:1 or 2:1 molar ratio (200 or 400 µM) at 70–80% confluence for 4 or 8 h. The overall amount of three relevant ceramides was measured by LC-MS/MS. Data were normalized to the total protein content of the samples and are shown as mean values ± S.D.; n ≥ 4; statistically significant differences: * *p* < 0.05 vs. BSA treated control; ^!^
*p* < 0.05 vs. palmitate treated samples.

	Overall Ceramide Content ^1^ (ng/mg Protein)	
Treatment	at 4 h	at 8 h
Control	171.1 ± 12.3 ^!^	149.3 ± 52.4 ^!^
Palmitate	765.3 ± 197.4 *	1 993.9 ± 100.5 *
cis-Oleate	134.8 ± 24.8 ^!^	88.7 ± 25.7 ^!^
trans-Elaidate	195.5 ± 18.0 ^!^	133.8 ± 38.7 ^!^
trans-Vaccenate	175.7 ± 23.0 ^!^	144.5 ± 51.1 ^!^
Palmitate + Oleate (4:1)	502.9 ± 125.1 *^,!^	946.4 ± 593.6 *^,!^
Palmitate + Oleate (2:1)	393.1 ± 90.9 ^!^	435.5 ± 85.3 ^!^
Palmitate + Elaidate (4:1)	591.4 ± 130.7 *	1 146.4 ± 204.7 *^,!^
Palmitate + Elaidate (2:1)	421.7 ± 95.0 ^!^	659.6 ± 59.4 ^!^
Palmitate + Vaccenate (4:1)	539.5 ± 137.8 *	1 173.4 ± 466.2 *^,!^
Palmitate + Vaccenate (2:1)	450.0 ± 113.9 *^,!^	760.0 ± 281.4 ^!^

^1^ Sum of the amounts of three types of ceramides: Cer 16:0, Cer 18:0 and Cer 18:1.

## Data Availability

Not applicable.

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
