# Peer review of "Different Metabolism and Toxicity of TRANS Fatty Acids, Elaidate and Vaccenate Compared to Cis-Oleate in HepG2 Cells"

_ijms, 2022, doi:10.3390/ijms23137298_

Round 1

Reviewer 1 Report

The manuscript represents an interesting approach to fatty acid supplementation and cell metabolism. However, there are several drawbacks from the manuscript in which an important amount of data was presented 

1.- The preparation of the fatty bound albumin is not done properly, incubation at 50 degrees is not suitable for the preparation of the BSA fatty acids see doi.org/10.2144/000114285

2.- The concentrations used are too high the range should've up to 100 microM concentration, concentrations above induce osmotic changes in cell culture. The use several of the compounds tested were already toxic at the lowest concentration. The electron microscopy analysis showed toxicity related to the high concentration used. Even though cell death is not obvious, cells are not completely viable. The authors used schemes as if they were generating foam cells.

Supplementary figure 7 does not seem to be done properly as the intensity of the bands is fainted for the tests as compared to GADPH as well as figure 6.

3.- The scheme of co supplementation was not done properly. The transport of saturated fatty acids generates carry over, see 

  • 10.1016/j.plefa.2010.02.029

4.-Most of the data on fatty acids and ceramides can be rescued only if the manuscript is devoted to toxic effects and possible mechanisms otherwise it not be accounted for. Moreover, more experiments should be performed to assess cell death.

5.- Figures S3 and S4 were not analyzed properly why there is a lack of increase in ceramide content with oleate treatment as expected.

6.- Minor issues, the manuscript lacks a proper structure, the abstract is vague, and the discussion does not provide enough arguments to explain the results. There are several important grammatical errors to correct. Finally, the reference citation is very poor.

Reviewer 2 Report

In this work, the metabolism, toxicity, and potential protective effect of elaidate and vaccinate fatty acids in human hepatocellular carcinoma cells were investigated. The cells were treated with palmitate, oleate, elaidate, and vaccinate fatty acids, alone or in combination with palmitate at different ratios (2:1 and 4:1). Lipotoxicity was induced by supplementing palmitate to the cell. The results showed that oleate has a significantly higher protective effect toward palmitate lipotoxicity (when co-supplemented ) than its trans-fatty acid elaidate and vaccinate. When supplemented alone, elaidate and vaccinate are slightly more toxic than oleate.

This is a very interesting work in a research field that deserves more attention. The manuscript is well organized. The communication is very clear, the results were properly presented and discussed. The discussion section is very thorough and well written. The figure's quality is good but could be improved. The same is true for the materials and methods section. Overall, this is good work, and I do recommend its publication after minor changes. Please find some comments and suggestions below:

  • Figures 1, 2, 3, S1, and S2 do not have error bars. Please add them to the figures.
  • Section 2.1 “Incorporation of fatty acids into the cells” – How do the authors ensure that figures 1 and 2 represent the fatty acid content incorporated into the cell? The cells were supplemented with high concentrations of free fatty acids. Is there a possibility that part of the fatty acids presents in the media or adsorbed on the cell surface was also included in this figure?  If so, please correct the text accordingly, by avoiding the use of the term “incorporated fatty acids in cells”.
  • The Total Fatty acid content was determined. However, I was surprised that the free fatty acid (FFA) content was not monitored. This would provide very interesting data for this work. For example, it would be interesting to observe if there were changes in the free fatty acid pool. Also, it would be interesting to understand how the supplementation of cis and trans fatty acids would affect that pool. Besides, monitoring the FFA content would also provide evidence to answer bullet point 2.  Is there any reason why these measurements were not considered?
  • The methods used in this work to determine TG and DG, do not have the level of detail to assign the carbon position of the fatty acids for those molecules. So, the data in figures 3 and 4 and along the text cannot be reported as it is. For example, when TAG(16:0/16:0/18:0) is reported, one will assume FA(16:0), FA(16:0), and FA(18:0) are assigned in the positions sn1, sn2, and sn3, respectively. Even though the FA composition is similar, that molecule is different than TG(16:0/18:0/16:0) or TG(18:0/16:0/16:0). In fact, these are three different TGs. When the sn position of the FA on a TG, DG, or phospholipid is not possible to assign, the molecules should be reported as TG(16:0_16:0_18:0). One will interpret this as a TAG containing two FA(16:0) and one FA(18:0). This is how the TG and DG data should be reported in the manuscript. Please use the following articles as a reference for the correct naming system 1, 2.
  • It is not clear how TG, DG, and Ceramide species were identified by the LC-MS/MS technique used. It would be useful to provide the full MRM list and a chromatogram showing the different retention times. Moreover, it is not clear how these molecules were quantified. Detailed information about the quantification method used should also be provided. Without this is very hard to evaluate and validate the quality of this data.
  • Bullet point  5) can also be applied to the GC-FID FA profile. Please provide a chromatogram with the different fatty acids of interest identified on it. It is also not clear how FAs were quantified. Please provide detailed information about the quantification method in the materials and methods section.
  • I am concerned if the FFA amounts used in this study are biologically relevant. It is known that a high concentration of free fatty acids will lead to lipotoxicity. The authors supplemented the cells with very high concentrations of free fatty acids. The concentration used in this work is justified based on other studies that used similar amounts. Even though other authors used high amounts of free fatty acids, that does not mean that those concentrations could be excessively high and induce an acute toxic response in the model. Did the authors make any dose-response study prior to deciding on the FFA concentration to use?

  • 1 Liebisch, G. et al. Shorthand notation for lipid structures derived from mass spectrometry. J Lipid Res 54, 1523-1530, doi:10.1194/jlr.M033506 (2013).
  • 2 Liebisch, G., Ekroos, K., Hermansson, M. & Ejsing, C. S. Reporting of lipidomics data should be standardized. Biochim Biophys Acta, doi:10.1016/j.bbalip.2017.02.013 (2017).

Reviewer 3 Report

The authors investigated the effect of oleic acid and its two trans isomers (elaidic and vaccenic acid) on palmitoleic acid toxicity in HepG2 cells. It is a nice study; however, I really miss the comparison of the results with other cell studies that have previously investigated the effects of these fatty acids on different cell types. Now it seems to me (according to this article) that this is a really new field of investigation, however, there are some really old ones investigating the differences in the effect of cis and trans (industrial and ruminant) isomers of C18:1.

It is good to put your research in medical context (proving that it is important from the medical / pathophysiological point of view), but it is an in vitro cell study, so you should mainly focus on results of other (previous) cell studies. Please compare your results with previous findings and highlight what was already known, what is new? Are there similarities or contradictions with previous findings from other research groups?

As your study is quite complex with many tables, figures, supplementary files and many aspects of the effect of different fatty acids on the HepG2 cells (cell viability, utilization of FAs, cellular TG, DG, ceramide content, co-supplementation, stress-related proteins, IC fat accumulation, ER-morphology), I would find helpful to sum your main findings in a short conclusion part.

Some minor comments:

 - The word “triacylglycerol” is more appropriate chemically than triglyceride. Please correct it. Similarly, please use “diacylglycerol” instead of diglyceride.

 - In row 469 you write that fatty acids are building blocks of phosphatides, what do you mean here? Do you mean phospholipids? Please clarify it and use the chemically proper name.

 - Row 471: The abbreviation TG has already been introduced (row 75), no need to reintroduce it.

Round 2

Reviewer 1 Report

Unfortunately, the authors did not take the comments seriously despite the fact that were documented accordingly. 

Author Response

Dear Reviewer 1,

Thank you.

Yours sincerely,

Miklós Csala

Reviewer 3 Report

The authors answered all my questions but I still believe that several cell studies are missing from the discussion part. I know that they are a bit different (investigating other cells or other, but still similar outcomes) compared to your study but I still believe that these former studies are essential for interpreting your results, presenting the literature and findings to date, and pointing out areas that have not yet been explored.

Some former studies that might be relevant for the discussion section (not the full list, just a small selection):

·        Woldseth et al (https://pubmed.ncbi.nlm.nih.gov/10088200/) investigated the metabolism and incorporation of different FAs (16:0 [PA], 18:0 [SA], 18:1c-9 [OA], 18:1t-9 [EA], 18:1t-7 [VA]) into PLs (PC and PE) in rat hepatocytes.

·        Hu et al (https://pubmed.ncbi.nlm.nih.gov/31895396/) studied the metabolism and incorporation of FAs into PLs in HUVEC cells after OA, cis-VA, trans-VA, EA incubation as well as the effect on the expression and secretion of PLA2 and different inflammatory cytokines (ICAM-1, VCAM-1, IL-6))

·        Valenzuela et al (https://pubmed.ncbi.nlm.nih.gov/34641380/) incubated endothelial cells (EA.hy926) with different FAs (OA, EA, trans-VA, cis-VA) and measured inflammatory mediators (MCP-1, ICAM-1, IL-6, IL-8) and genes (NFκB1, TLR-4, PTGS2), cell adhesion and differences in cell surface expression of ICAM-1.

·        Silva et al (https://pubmed.ncbi.nlm.nih.gov/28315997/) also studied the effect of different trans isomers (tPA, VA, EA) on inflammatory gene expressions and PGE2 excretion in HUVEC and HepG2 cells.

·        Li et al (https://pubmed.ncbi.nlm.nih.gov/29367652/) also investigated the difference between EA and VA on pro-inflammatory factor expressions (ICAM-1, VCAM-1, IL-6) and MAPK signalling pathway in HUVEC cells.

·        Vargas-Bello-Perez et al (https://pubmed.ncbi.nlm.nih.gov/31193913/) treated bovine mammary cells (MAC-T) with PA, SA, OA, EA, VA and investigated the differences in cytosolic TG accumulation.

·        Krogager et al (https://pubmed.ncbi.nlm.nih.gov/26628894/) treated HEP-G2 cells with different FAs (OA, cis-VA, trans-VA, EA) and investigated the uptake and incorporation of these FAs into TG.

·        A quite similar study about incorporation of OA, EA, cis- and trans-VA into TGs and PLs in rat liver cells: https://pubmed.ncbi.nlm.nih.gov/21614647/

·        Van Greevenbroek et al (https://pubmed.ncbi.nlm.nih.gov/9734731/) studied the difference in cell FA composition and lipoprotein secretion in Caco-2 cells after PA, SA, OA and EA treatment.

·        Dashti et al (https://pubmed.ncbi.nlm.nih.gov/11108731/) incubated HepG2 cells with OA, EA and PA and measured their incorporation into TGs and other lipoproteins.

·        Sauvat et al (https://pubmed.ncbi.nlm.nih.gov/29606629/) compared the different effect of saturated (PA), cis- and trans-unsaturated FAs (OA, EA) on the autophagy in cultured cells and in in vivo models also.

·        Ahmed et al (https://pubmed.ncbi.nlm.nih.gov/28340693/) investigated the cellular lipid content and intracellular lipid droplets after OA, PA and EA incubation.

From this list you can see that the first papers were published as early as 1998, so the potential effects of trans isomers have already been investigated at the cellular level by several research groups using several different cell lines. So please include the relevant ones into the discussion part and elucidate what is known and what is new in this study.

Author Response

Dear Reviewer #3,

Thank you for helping to further improve our manuscript by specifying some articles to be considered for inclusion in the Discussion.

We have extended the Discussion with 10 studies [30-39], and did our best to highlight the novelty of our own work. The changes are located at rows 593-640. Most importantly, uptake of TFAs and their incorporation into TGs and membrane phospholipids have been demonstrated in various cells, but their effects on ceramide and DG accumulation have not been investigated, nor have they been correlated with ER stress, JNK activation and apoptosis in single treatments and in combination with palmitate.

We hope that you will find our manuscript acceptable in the present form.

Thank you once again for your help and support.

Yours sincerely,

Miklós Csala

Round 3

Reviewer 3 Report

I accept the answers to my questions and thank you for adding some previous cell studies about TFAs.